# YgfY Contributes to Stress Tolerance in *Shewanella oneidensis* Neither as an Antitoxin Nor as a Flavinylation Factor of Succinate Dehydrogenase

**DOI:** 10.3390/microorganisms9112316

**Published:** 2021-11-09

**Authors:** Ming-Xing Zhang, Kai-Li Zheng, Ai-Guo Tang, Xiao-Xia Hu, Xin-Xin Guo, Chao Wu, Yuan-Yuan Cheng

**Affiliations:** 1School of Life Sciences, Anhui University, Hefei 230602, China; zstar94@163.com (M.-X.Z.); d19301054@stu.ahu.edu.cn (K.-L.Z.); d19201030@stu.ahu.edu.cn (A.-G.T.); d20201038@stu.ahu.edu.cn (X.-X.H.); guoxx2017@yeah.net (X.-X.G.); 2Anhui Provincial Engineering Technology Research Center of Microorganisms and Biocatalysis, Anhui University, Hefei 230602, China; 3School of Resources and Environment, Anhui University, Hefei 230602, China; benny928@mail.ustc.edu.cn

**Keywords:** YgfY, *Shewanella*, toxin–antitoxin, succinate dehydrogenase, flavinylation

## Abstract

YgfY(SdhE/CptB) is highly conserved while has controversial functions in bacteria. It works as an antitoxin and composes a type IV toxin–antitoxin system with YgfX(CptA) typically in *Escherichia coli*, while functions as an flavinylation factor of succinate dehydrogenase and fumarate reductase typically in *Serratia* sp. In this study, we report the contribution of YgfY in *Shewanella oneidensis* MR-1 to tolerance of low temperature and nitrite. YgfY deficiency causes several growth defects of *S. oneidensis* MR-1 at low temperature, while YgfX do not cause a growth defect or morphological change of *S. oneidensis* MR1-1 and *E. coli*. YgfY do not interact with FtsZ and MreB nor with YgfX examined by bacterial two-hybrid assay. YgfY effect on growth under low temperature is not attributed to succinate dehydrogenase (SDH) because a mutant without SDH grows comparably with the wild-type strain in the presence of succinate. The *ygfY* mutant shows impaired tolerance to nitrite. Transcription of nitrite reductase and most ribosome proteins is significantly decreased in the *ygfY* mutant, which is consistent with the phenotypes detected above. Effects of YgfY on growth and nitrite tolerance are closely related to the RGXXE motif in YgfY. In summary, this study demonstrates pleiotropic impacts of YgfY in *S. oneidensis* MR-1, and sheds a light on the physiological versatility of YgfY in bacteria.

## 1. Introduction

*Shewanella* species are emerging environmental bacteria with unique and versatile respiration, therefore playing a role in the biogeochemical cycle of metals [1,2]. In their favored niches, *Shewanella* species might confront stresses, such as heavy metals, toxic chemicals, high pressure and cold temperature [3]. However, the current knowledge regarding stress tolerance in the *Shewanella* species is still limited.

Physiological functions of bacterial toxin–antitoxin (TA) systems are commonly investigated in a number of laboratory and clinical isolates. In contrast, their physiological significance in environmental bacteria is relatively obscure. Recent studies have revealed a novel TA system in *S. oneidensis* MR-1 that plays a role in cell motility [4].

TA systems include a toxin protein that hinders cell growth, and an antitoxin as a protein or an RNA that protects cells from toxins via various mechanisms. Up to date, six types of TA system have been reported [5]. The type IV TA system is different from all the other TA systems in which antitoxins directly target the toxin. Instead, the antitoxin protein in type IV system protects cells from the toxin through interacting with and stabilizing the targets of the toxin. All type IV TA systems reported so far are involved in cell division through interfering functions of cytoskeleton proteins FtsZ and MreB. CptB(YgfY)/CptA(YgfX) and CbtB/CbtA are the first two identified type IV TA systems. Both toxins of CptA and CbtA inhibit the cell division through interacting with cytoskeleton proteins, FtsZ and MreB [6,7]. CbtB antagonizes the CbtA toxicity by enhancing the bundling of cytoskeletal polymers of MreB and FtsZ in *Escherichia coli* [6]. YkfI and YpjF, CbtA homologous in *E. coli* also inhibit cell growth and lead to morphological changes of cells by interacting with FtsZ and MreB [8].

Interestingly, although highly conserved in diverse bacteria, CptB(YgfY) homologs show the other totally irrelevant activity. SdhE, a CptB(YgfY) homolog in *Serratia* sp. and *Acetobacter pasteurianus,* works as a flavinylation factor of succinate dehydrogenase (SDH) [9,10]. SdhE is even involved in flavinylation of fumarate reductase in *Serratia* sp., demonstrating the broadness of the target list of CptB/YgfY/SdhE [11].

This study aims to reveal the physiological function of YgfY/YgfX and explores the underlying mechanism in *S. oneidensis* MR-1. Our results demonstrate that YgfY has pleiotropic impacts in *S. oneidensis* MR-1 including adaptive growth at low temperature and tolerance to nitrite. However, these impacts of YgfY were not attributed to functions as an antitoxin nor as a flavinylation factor of SDH. Lines of evidence in this study suggest a novel mechanism of YgfY function in stress tolerance of *Shewanella* species.

## 2. Materials and Methods

### 2.1. Bacteria and Growth Conditions

Bacterial strains were cultured in Lysogeny Broth (LB) medium or mineral medium as reported previously [12]. Strains of *E. coli* were cultured at 37 °C, and those of *S. oneidensis* MR-1 were cultured at 30 °C or 16 °C as indicated. Chemicals were added when needed at the following concentrations: 100 μg/mL diaminopimelic acid, 50 μg/mL kanamycin, 100 μg/mL ampicillin, and 20 μg/mL gentamycin for strains of *E. coli*; 10 μg/mL gentamycin for strains of *S. oneidensis* MR-1.

### 2.2. Construction of Strains and Plasmids

The construction of transposon library and the location of transposon insertion were conducted as described previously [13], except that *S. oneidensis* MR-1 was used as the parent strain of the library.

Mutants with an in-frame deletion of desired genes were constructed as described previously [14]. Briefly, flanking regions of a target gene were amplified and ligated with digested suicide vector pRE112 [15] using ClonExpress MultiS One Step Cloning Kit (Vazyme, Nanjing, China). The ligation products were transformed into *E. coli* SM10 and verified by PCR. Verified plasmids were extracted and transformed into *E. coli* WM3064 [16] and then introduced into strains of *S. oneidensis* MR-1 by conjugation. After two rounds of selection, a mutant was obtained with the deletion of a target gene and confirmed by sequencing.

For complementation, *ygfY* from *S. oneidensis* MR-1 was cloned into pBBP1 [9] using ClonExpress II One Step Cloning Kit (Vazyme, Nanjing, China). pBBP1 contains a *cymA* promoter that has constitutive activity and drives expression of desired genes in *S. oneidensis* MR-1. Generated plasmids were sequentially introduced into *E. coli* SM10, *E. coli* WM3064 and strains of *S. oneidensis* MR-1. For protein expression in *E coli* BL21(DE3), *ygfX* was cloned into pBAD24 [17] to construct pBAD_*ygfX* and *ygfY* was cloned into pET-28a(+) to construct pET_*ygfY*. pBAD_*ygfX* and pET_*ygfY* were co-transferred into *E. coli* BL21(DE3).

Site-directed mutation was conducted to construct pBBP1_*ygfY*^G16R/E19A^ as described previously [13]. Briefly, primers with desired substitution at the 5′ ends of joint primers sites were designed and used for overlapping PCR. PCR products were ligated into digested pBBP1 using ClonExpress II One Step Cloning Kit (Vazyme, Nanjing, China). All generated plasmids were verified by sequencing before phenotype tests. All strains and plasmids used in this study were listed in Appendix A and all primers were listed in Appendix A.

### 2.3. Overexpression of YgfY and YgfX in E. coli

*E. coli* BL21(DE3) bearing pBAD_*ygfX* and pET_*ygfY* was cultured in LB overnight and then diluted in fresh LB to 0.01 of OD_600_ for sub-cultivation. Inducers (isopropyl β-D-thiogalactoside (IPTG) and L-arabinose) or inhibitor (glucose) were added when subcultures grew to 0.2 of OD_600_. IPTG and/or L-arabinose were added to 1 mM of final concentration to induce expression of YgfY and YgfX, alone or simultaneously. Glucose was added to 1 mM of final concentration to suppress leaking expression both of YgfY and YgfX. Aliquots of subcultures were withdrawn to measure OD_600_ and observe cell morphology using a microscope Axio Scope A1 (Carl Zeiss Microscopy, LLC., New York, NY, USA).

### 2.4. Bacterial Two-Hybrid Assay

The bacterial two-hybrid system based on adenylate cyclase reconstitution in *E. coli* was adopted to detect the interaction of YgfX with other proteins in vivo (Euromedex, Souffelweyersheim, France). *ygfX* was amplified and cloned into pKNT25 and pKT25 to express a chimeric protein that fused T25 fragment of adenylate cyclase to the N-terminal and C-terminal of YgfY, respectively. *ftsZ*, *mreB*, and *ygfX* were cloned into pUT18 and pUT18C. Detection of protein–protein interaction in vivo was conducted according to manufacturer’s instruction (Euromedex, Souffelweyersheim, France). Overnight cultures of *E. coli* BTH101 bearing two plasmids were used to determine β-galactosidase activity in cells.

### 2.5. RNA Isolation and Sequencing

Overnight cultures in LB were diluted in fresh LB to 0.1 of OD_600_ and then cultivated at 16 °C for 12 h. Four biological replicates of subcultures were mixed and collected by centrifugation (12,000× *g*, 4 °C, 1 min). Collected cells were subjected to extraction of total RNA using the Trizol Reagent (Takara Biotechnology, Beijing, China) according to the manufacturer’s instruction. Total RNA was further purified using RNA clean kit (BioTeke, Beijing, China). The concentration, quality, and integrity of total RNA were determined using a NanoDrop spectrophotometer (Thermo Fisher Scientific, Waltham, MA, USA) and an Agilent 2100 bioanalyzer (Agilent Technologies, Waltham, CA, USA). Qualified total RNA was used to construct a sequencing library that was subsequently sequenced on a Hiseq platform (Illumina, New York, NY, USA), which was performed by Shanghai Personal Biotechnology Co., Ltd.

The data of RNA sequencing (RNA-Seq) were deposited to GOE. The accession numbers have not yet been obtained but will be provided before acceptance of this manuscript.

### 2.6. qRT-PCR Assay

cDNA was synthesized using 500 ng of qualified total RNA and PrimeScript RT reagent kit with gDNA eraser (Takara Biotechnology, Beijing, China). The qRT-PCR was conducted using the SYBR Premix Ex Taq kit (Takara Biotechnology, Beijing, China) and Lightcycler 96 (Roche, Mannheim, Germany). The *gyrB* was used as the reference gene. The relative expression value of target genes was obtained from three determinations with normalization against the reference gene using the method of 2^−ΔΔCt^.

### 2.7. Assay of Transcription Capability Using Click Chemistry

To detect the transcription capability, a method of click chemistry using 5-ethynyluridine and an azide-modified fluorophore was adopted [10]. Overnight cultures of WT and ∆*ygfY* were diluted into fresh LB to 0.01 of OD_600_, and then cultured at 30 °C for 3 h. Then, cultures were collected, concentrated for 25 folds, and added with EU (Molecular Probes, Inc., Eugene, OR, USA) to a final concentration of 0.5 mM. Cultures were allowed to grow for another 2 h at 30 °C. After that, cells were collected, fixed with 4% formaldehyde, and neutralized with 2 mg/mL glycine. After washed with phosphate-buffered saline (PBS; 137 mM NaCl, 2.7 mM KCl, 10 mM Na_2_HPO_4_, 1.8 mM KH_2_PO_4_, pH 7.2), cells were incubated in 0.5% Triton X-100/PBS for 10 min. After washed with PBS, cells were stained by 2 μM azide-modified Alexa Fluor 488 (Molecular Probes, Inc., Eugene, OR, USA) in Click-iT cell reaction buffer (Molecular Probes, Inc., Eugene, OR, USA) for 30 min at room temperature. Stained cells were washed with PBS, and then observed using a microscope Axio Scope A1 (Zeiss, Jena, Germany).

### 2.8. Assay of Nitrite Tolerance

To determine the minimal inhibition concentration (MIC) of nitrite, *S. oneidensis* MR-1 was cultured at 30 °C for 16 h and then diluted in fresh LB to 0.01 of OD_600_. Nitrite was added into those diluted cultures to final concentrations of 0 to 1000 mM. Diluted cultures were cultivated at 30 °C for 24 h.

The nitrite tolerance of Δ*ygfY* and wild-type (WT) was determined by plating nitrite-treated cells on LB agar plates and counting colony-forming units (CFU). In detail, strains were cultured in LB at 30 °C for 16 h and then diluted in fresh LB to 0.01 of OD_600_. Diluted cultures were subcultivated to 0.2–0.3 of OD_600_ before nitrite of 470 mM (15 × MIC) was added. At indicated time points, aliquots of nitrite-treated subcultures were withdrawn, washed with 0.9% NaCl two times to remove remained nitrite. Washed cultures were serially diluted in 0.9% NaCl and spotted on LB agar plates. Cells on plates were cultivated at 30 °C for 24 h before counting CFU.

### 2.9. Assay of Tolerance to Heat Shock

Strains were cultured in LB at 30 °C for 16 h and then diluted in fresh LB to 0.01 of OD_600_. Diluted cultures were subcultivated to 0.2–0.3 of OD_600_ at 30 °C, and then incubated at 42 °C. At indicated time points, aliquots of heat-shocked subcultures were withdrawn, diluted with 0.9% NaCl, and spotted on LB agar plates. If needed, 100 U/μL catalase solution of 5 μL was overlayed on culture spots. Cells on plates were cultivated at 30 °C for 24 h before counting CFU.

### 2.10. Quantification of Succinate

Strains were cultured in LB overnight. Cells were collected by centrifugation (3000× *g*, 10 min) for 1 min and washed three times with a mineral medium (MM) [12]. After that, cells were resuspended in MM containing 10 mM succinate to final density of 2.5 of OD_600_. These cultures were incubated at 30 °C with vigorous shaking, and aliquots were withdrawn at indicated time points. Aliquots of samples were centrifugated at 12,000× *g* for 10 min and treated with 0.22 μm filters. Succinate remained in MM was quantified using Agilent 1260 liquid chromatography (Agilent Technologies Inc., Santa Clara, CA, USA) equipped with an Aminex HPX-87H column (Bio-Rad Laboratories, Inc., Hercules, CA, USA). H_2_SO_4_ of 5 mM was used as a running buffer at a flow rate of 0.5 mL/min. The standard calibration curve of succinate was obtained using MM containing succinate over range of 1–50 mM.

## 3. Results

### 3.1. YgfY Is Required for Normal Growth of S. oneidensis MR-1

A transposon library of *S. oneidensis* MR-1 was constructed to screen genes that contribute to tolerance to various stresses. We accidently found a colony showing smaller size than WT (data not shown). The transposon in this mutant inserted into a putative two-gene operon containing *ygfY* (SO_1339) and *ygfX* (SO_1340). The *ygfY* and *ygfX* are predicted as an operon coding a type IV TA system, in which *ygfY* encodes an antitoxin and *ygfX* encodes a toxin.

To confirm the effect of YgfY and YgfX on cell growth, mutants with in-frame deletion of *ygfX* and *ygfYX* were constructed. Intriguingly, we also successfully constructed a *ygfY* mutant that is theoretically inaccessible if YgfY was a cognate antitoxin of YgfX. Δ*ygfY* and Δ*ygfYX* rather than *ΔygfX* showed defect in growth at 30 °C (Appendix A). Δ*ygfY* and Δ*ygfYX* showed more severe defect of growth when cultivated at 16 °C (Figure 1A). Growth was quantified by the optimal density of cultures at 600 nm that could be affected by the cell density as well as the cell size. The cell density of cultures was further determined by counting colony forming units (CFU). After cultivated at 16 °C for 48 h, cultures of Δ*ygfY* reached to (2.02 ± 0.30) × 10^9^ CFU/mL and those of WT reached to (3.62 ± 0.15) × 10^9^ CFU/mL. The CFU of Δ*ygfY* cultures was about 56% of those of WT, which was consistent with the result of optimal density measurement. Meanwhile, cells in these cultures were observed by microscopy, and no morphological difference was observed among Δ*ygfY*, Δ*ygfX*, Δ*ygfYX* and WT (Appendix A). Complementation of *ygfY* completely restored the growth of Δ*ygfY* and Δ*ygfY*X to a similar level of WT (Figure 1B). Expression of *ygfY* from its native promoter also restored the growth of Δ*ygfY* and Δ*ygfY*X to a similar level of WT (Appendix A).

These results indicated that YgfY rather than YgfX positively affected the growth and did not support YgfY/YgfX in *S. oneidensis* MR-1 as a TA system, because ∆*ygfYX* lacking both YgfY and YgfX should show growth recovery compared with ∆*ygfY* if YgfX functions as a toxin and YgfY as a cognate antitoxin.

### 3.2. YgfY and YgfX Do Not Fulfil the Criteria of Type IV TA System

Both *ygfY* and *ygfX* are supposed to compose a two-gene operon in which the start codon of *ygfY* is 37 bp away from the stop codon of *ygfX* (Figure 2A). The RT-PCR detected the transcription of the intergenic region between *ygfY* and *ygfX*, indicating that they belong to the same operon (Figure 2B). The similarity of amino acid sequences between YgfY homologs in *S. oneidensis* MR-1 and *E. coli* MG1655 was 62.5%, and that between YgfX homologs in two strains was 34.8%. EMBOSS Needle was used to analyze the similarity of amino acid sequences.

Hetero-expression of toxins in *E. coli* commonly causes growth defect. To further examine physiological function of YgfY and YgfX, they were overexpressed in *E. coli* BL21(DE3). Expression of YgfY was driven by an IPTG-induced promoter in pET-28a(+) and that of YgfX was driven by an arabinose-induced promoter in pBAD24. As the blank controls, cultures of *E. coli* BL21(DE3) in the early exponential phase was added with 1 mM glucose as an inhibitor to minimize leaking expression of YgfY and YgfX. After addition of inducers or glucose for 4 h, cultures expressing either YgfX or YgfY showed growth defect, and those expressing both YgfX and YgfX showed a more severe growth defect compared with the glucose-added cultures (Figure 3A). This result did not support YgfX and YgfY functioning as a toxin and cognate antitoxin, respectively. Otherwise, expression of YgfY should rescue the growth defect caused by YgfX expression. Many proteins become toxic when highly overexpressed as published previously [18], which probably explains the growth defect observed upon overexpression of YgfX or YgfY. Overexpression of toxin YgfX (CptA) from *E. coli* in its native host causes change of cell morphology to the lemon shape [7]. Therefore, the cell morphology of *E. coli* BL21(DE3) after expressing YgfY and/or YgfX was observed. No morphological change was observed for cells expressing YgfX or YgfY or both (Figure 3B). No phenotype was observed in corresponding deletion mutants and overexpression strains, showing that YgfY and YgfX did not follow one of criteria of the type IV TA system.

YgfX (CptA) in *E. coli* hinders cell division through interacting with cytoskeleton proteins, FtsZ and MreB [7]. CbtA, the toxin in another type VI TA system, also interacts with FtsZ and MreB to hinder division of *E. coli* cells [6]. The interaction of YgfX with FtsZ and MreB was examined using bacterial two hybridization system. Compared to the negative and positive controls, no obvious interaction was detected either between YgfX and FtsZ nor between YgfX and MreB (Figure 4A,B). In addition, no interaction between YgfY and YgfX was detected (Figure 4C).

### 3.3. YgfY Does Not Attribute to Succinate Catabolism

YgfY in *S. oneidensis* MR-1 shares 60.2% similarity with its homolog SdhE in *Serratia* sp. SdhE in *Serratia* sp. functions as a flavinylation factor of SDH and fumarate reductase [11,19]. We examined whether growth defect of Δ*ygfY* was attributed to succinate metabolism. *S. oneidensis* MR-1 encodes SDH and can consume succinate. However, it cannot use succinate as a sole carbon source to support growth [20]. Therefore, we first compared the growth of Δ*ygfY*, Δ*sdh* and WT in the mineral medium supplemented with lactate and succinate. Lactate was added as a carbon source. There was no obvious difference in growth between Δ*sdh* and WT, while Δ*ygfY* showed a growth defect compared with WT in the mineral medium (Figure 5A). Then, we examined succinate consumption by resting cells in the mineral medium. After incubation for 12 h, Δ*ygfY* showed a comparable ability to consume succinate with WT, while Δ*sdhE* showed an obvious defect in succinate consumption (Figure 5B).

SdhE in *Serratia* sp. possesses an RGXXE motif that is essential for its activity. SdhE variants with substitution mutation of G16R or E19A in RGXXE motif are nonfunctional [19]. YgfY in *S. oneidensis* MR-1 also possess this motif (Figure 6A). Surprisingly, the RGXXE motif is highly conserved in SDH flavinylation factor in diverse organisms such as *Saccharomyces cerevisiae*, *Arabidopsis thaliana* and *Homo sapiens*. Mutation of G in the RGXXE motif of this factor is closely related to hereditary paraganglioma [21]. Therefore, we examined whether the RGXXE motif is also essential for the activity of YgfY in *S. oneidensis* MR-1. A YgfY mutant (YgfY^G16R/E19A^) was constructed with substitution of glycine and glutamic acid in RGXXE motif by arginine and alanine, respectively. YgfY^G16R/E19A^ was readily expressed in Δ*ygfY* (Appendix A), while did not rescue the growth defect of Δ*ygfY* (Figure 6B). These results confirmed that RGXXE motif was also essential for the activity of YgfY in S. oneidensis MR-1. These results indicated that the growth effect of YgfY should attribute to other activity rather than SDH flavinylation.

### 3.4. YgfY Contributes to Capability of Transcription and Translation in S. oneidensis MR-1

To explore the underlying mechanism of YgfY effect on growth in *S. oneidensis* MR-1, the transcriptomes of Δ*ygfY* and WT were compared. Δ*ygfY* and WT were cultured to the early exponential phase and subjected to analysis of RNA sequencing. Some housekeeping genes involved in transcription, translation, cell wall synthesis, and cell division were down-regulated (Appendix A) in Δ*ygfY*. Among them, genes encoding ribosome proteins were noticeable. Fifty out of fifty-four genes for ribosome proteins were down-regulated for more than two folds (Appendix A). The result of qRT-PCR confirmed the decrease in transcription of representative genes (Figure 7). The rate of ribosome formation is an integral part of the regulation of cell growth [22]. The decreased transcription of ribosome proteins in Δ*ygfY* is consistent with growth defect of this mutant.

The transcription capability of Δ*ygfY* was compared with WT using EU-based click chemistry [10]. EU is an uracil analog, and is incorporated into nascent RNA after addition. A weaker signal was detected in Δ*ygfY* than in WT after addition of EU for 2 h (Figure 8), indicating that the transcription capability was impaired in Δ*ygfY*.

Then, the translation capacity was compared by examining the tolerance to lethal stresses that cause global protein misfolding. Previous research reports that bactericidal aminoglycosides cause mistranslation, misfolding and aggregation of nascent proteins, which correlates to reactive oxygen species (ROS)A production and consequent cell death [23,24]. Inhibitors of protein synthesis, such as chloramphenicol, block ROS-related death of bacterial cells stressed by bactericidal antibiotics [25]. Therefore, we reasoned that if YgfY deficiency causes the lowered capacity of translation, Δ*ygfY* should show an increase in the tolerance to these stresses. The heat shock causes protein misfolding and aggregation [26], and there is a numerical correlation between the protein denature and the death of bacterial cells after the heat shock [27]. Consistent with our presumption, Δ*ygfY* showed an increased tolerance to heat killing than WT (Figure 9A). Moreover, catalase supplementation to quench ROS greatly improved the survival rate of WT, while not so much for that of Δ*ygfY* (Figure 9B).

In addition to house-keeping genes, transcription of some functional genes was also changed, such as two genes (*nrfA* and *nsrR*) involved in nitrite reduction. Transcription of *nrfA* and *nsrR* were down-regulated in Δ*ygfY* (Appendix A), which was also confirmed by qRT-PCR (Figure 6). The nitrite tolerance of Δ*ygfY* was examined next. When exposed to nitrite at a concentration of 15× MIC, Δ*ygfY* showed significantly more impaired tolerance to nitrite than WT, and the impairment was readily reversed by expression of YgfY but not YgfY^G16R/E19A^ (Figure 10).

## 4. Discussion

YgfXY homologs in *E. coli* and *Acinetobacter baumannii* were previously found to form a type IV TA system pair, which could not be confirmed later in *Serratia* sp. by different groups [28,29]. Similarly, we are unable to reproduce the toxic activity of YgfXY homologs in *S. oneidensis* MR-1. On the other hand, YgfY homologs in *E. coli*, *Serratia* sp. and *Acetobacter pasteurianus* show the activity as assembly factor for flavinylation of SDH [30,31]. In this study, we reveal that YgfY has pleiotropic impacts in physiology of *S. oneidensis* MR-1, but neither as an antitoxin nor as a flavinylation factor of SDH.

Genome annotation proposes YgfY and YgfX in *S. oneidensis* MR-1 as a TA system presumably for two reasons: (i) YgfY shares high similarity with the antitoxin CptB in *E. coli*, and (ii) *ygfY* locates adjacent to and in front of *ygfX* in the genome, which might compose a two-gene operon. Antitoxin CptB and toxin CptA in *E. coli* compose a type IV TA system in which CptA hinders cell division and causes the morphological change of cells through interacting with FtsZ and MreB, while CptB antagonizes CptA through stabilizing FtsZ and MreB [7]. Interaction of CptB with FtsZ and MreB causes morphological change of *E. coli* cells. Although YgfY and YgfX are co-transcribed and deficiency of YgfY causes growth defect of *S. oneidensis* MR-1 (Figure 1), other lines of evidence did not support that YgfY and YgfX function as a type IV TA system. Firstly, Δ*ygfYX,* deficient of both YgfX and YgfY, shows the same growth defect as Δ*ygfY* (Figure 1). If YgfX functioned as a toxin and was antagonized by YgfY, deletion of *ygfX* in Δ*ygfY* should rescue the growth defect of Δ*ygfY*. Secondly, overexpression of YgfX in *E. coli* does not cause morphological change of cells (Figure 3). Thirdly, YgfX does not interact with FtsZ or MtrB (Figure 4). Overall, YgfY and YgfX do not fulfill the criteria of a type IV TA system.

The severe growth defect of Δ*ygfY* also cannot be attributed to SDH because Δ*sdh* shows an unimpaired growth (Figure 5), although interaction between YgfY and SDH cannot be excluded. Δ*ygfY* shows a global change in transcriptome compared with WT (Figure 6, Appendix A). It is very unlikely that YgfY directly regulates transcription for the absence of any known DNA-binding motifs. Hence, transcriptome change for YgfY deficiency is more likely a feedback regulation of cells through other signals and regulators. Interestingly, SdhE in *Serratia* sp. also demonstrates pleiotropic impacts. Except for inability to use succinate, a mutant of *Serratia* sp. without *sdhE* shows a decrease in the transcription of an operon responding to synthesize an antibiotic prodigiosin [30], which is also hardly to be explained by SdhE functioning as a flavinylation factor of SDH.

Results from site mutation of YgfY provides some hints about the mechanism underlying pleiotropic impacts of YgfY. YgfY possesses a conserved RGXXE motif in flavinylation factor of SDH that is surprisingly conserved in all three kingdoms (Figure 5). Mutation of G16 in RGXXE motif abolishes the in vivo activity of YgfY in *S. oneidensis* MR-1 (Figure 5 and Figure 7). Structure investigations show that Gly in the RGXXE motif of the flavinylation factor of SDH plays the exact same role in all investigated organisms crossing three kingdoms. Gly in RGXXE motif forms a critical hydrogen with SdhA in SDH in *E. coli* and humans [32,33]. Moreover, Gly in RGXXE of SdhE(CptB) can also form a hydrogen bond with FrdA in the fumarate reductase complex and is indispensable for FrdA flavinylation in *E. coli* [34]. Based on those investigations, we propose that YgfY likely functions as a flavinylation factor as do its homologs in diverse organisms, while it has unidentified targets that eventually impose effects on growth and nitrite tolerance. Identification of those targets are on the way to deepen our view of this family of flavinylation factors.

## Figures and Tables

**Figure 1 microorganisms-09-02316-f001:**
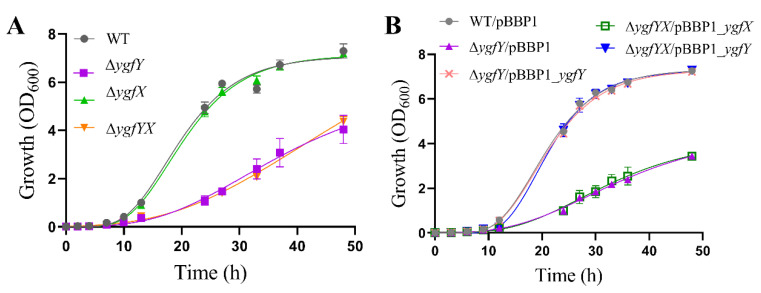
YgfY but not YgfX affected growth of *S. oneidensis* MR-1. (**A**) Growth of WT and mutants. (**B**) Complementation of YgfY by expressing it from a constitutive promoter in pBBP1. All strains were cultured in LB at 16 °C under aerobic condition. The data are the mean ± SD (*n* = 3). Error bars for some data points were too small to be shown.

**Figure 2 microorganisms-09-02316-f002:**
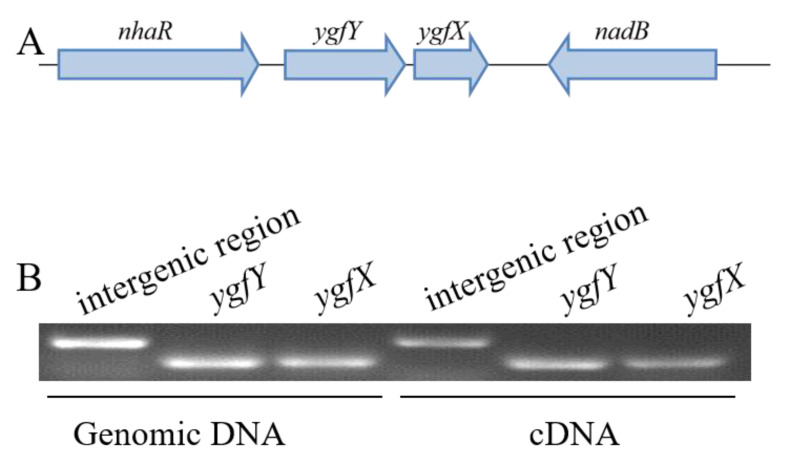
The *ygfY* located in an operon with *ygfX*. (**A**) Schematic of arrangement of *ygfY*, *ygfX* and their neighbor genes in genome of *S. oneidensis* MR-1. (**B**) Transcription of *ygfY* and *ygfX* and intergenic region between them.

**Figure 3 microorganisms-09-02316-f003:**
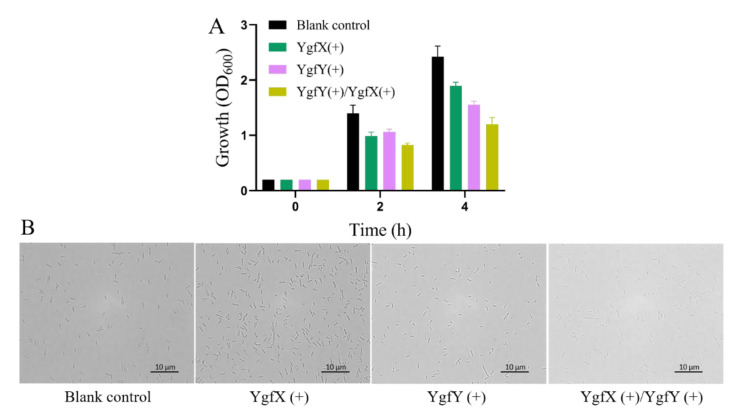
Hetero-expression of YgfY in *E. coli* BL21(DE3). (**A**) Growth of *E. coli* BL21(DE3) expressing YgfY and/or YgfX under induction. Cultures of *E. coli* BL21 bearing pBAD24_*ygfX* and pET_*ygfY* were grown to early exponential phase before the addition of inducers or glucose. The data are the mean ± SD (*n* = 3). (**B**) Cell morphology observed after induction for 4 h.

**Figure 4 microorganisms-09-02316-f004:**
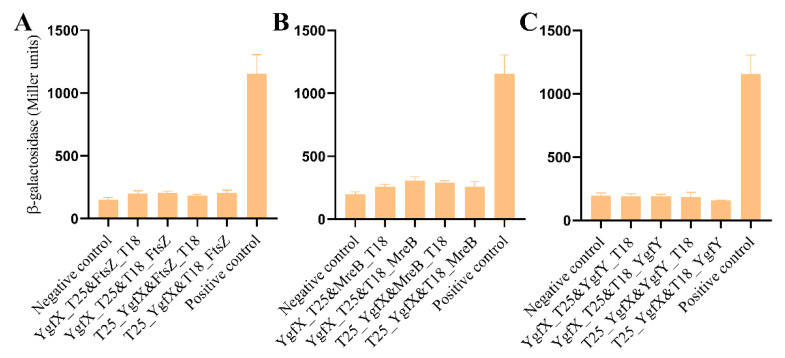
YgfX did not interact with (**A**) MreB, (**B**) FtsZ, and (**C**) YgfY. *E. coli* BTH101 expressing YgfX and the other protein simultaneously was examined for β-galactosidase activity in cells. The data are the mean ± SD (*n* = 3).

**Figure 5 microorganisms-09-02316-f005:**
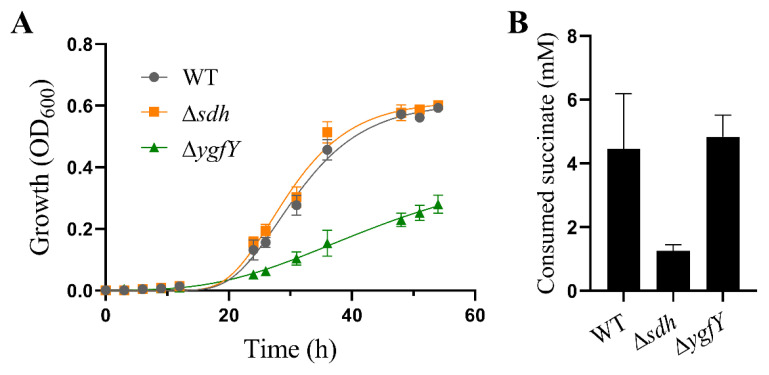
YgfY did not involve in succinate metabolism. (**A**) Growth in the mineral medium with 50 mM lactate and 10 mM succinate. (**B**) Succinate consumption by resting cells for 12 h in the mineral medium containing 10 mM succinate. The data are the mean ± SD (*n* = 3). Error bars for some data points were too small to be shown.

**Figure 6 microorganisms-09-02316-f006:**
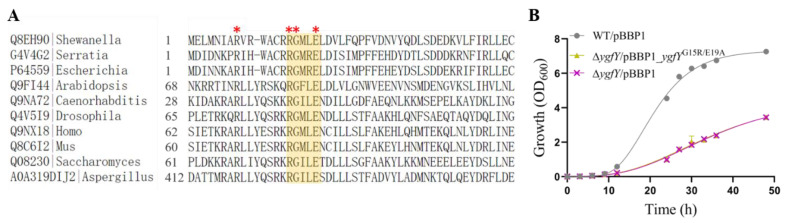
Conserved sites essential for YgfY activity. (**A**) Amino acid alignment of succinate dehydrogenase flavinylation factors and YgfY homologs. Entry of proteins in Uniport and family of organisms are indicated. Red asterisk (*) indicate conserved amino acid residues and RGXXE motif is highlighted with a yellow box. (**B**) Growth of ∆*ygfY* complemented with YgfY mutant (YgfY^G16R/E19A^). The data are the mean ± SD (*n* = 3). Error bars for some data points were too small to be shown.

**Figure 7 microorganisms-09-02316-f007:**
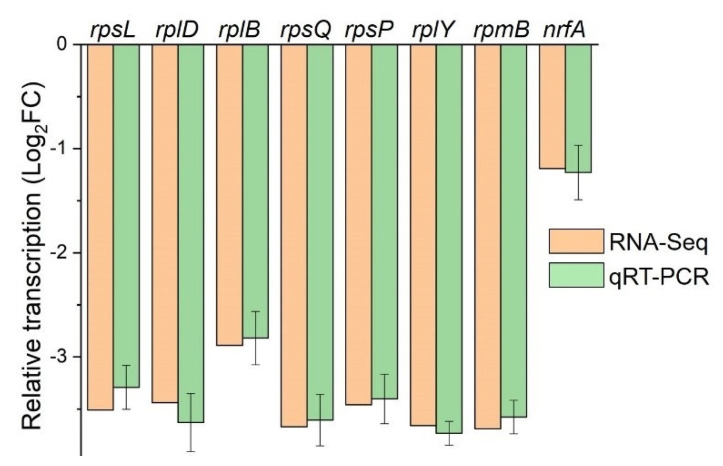
Transcription of representative genes that were down-regulated in Δ*ygfY*. Δ*ygfY* and WT were cultured at 16 °C for 12 h for transcriptome analysis (RAN-Seq). Transcription (log2(fold change)) of genes was examined using qRT-PCR and compared with RNA-seq data. Error bars indicate standard deviations of results from three biological replicates.

**Figure 8 microorganisms-09-02316-f008:**
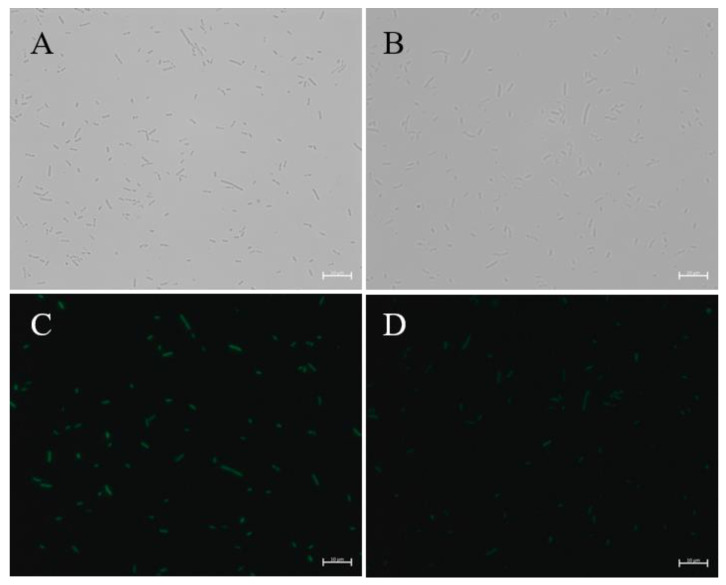
RNA synthesis in WT and Δ*ygfY*. Cultures at the early exponential phase were exposed to EU. Nascent RNA incorporated with EU was detected by azide-modified Alexa Fluor 488 under a microscope. (**A**,**C**) Shows cells of WT under brightfield and fluorescence. (**B**,**D**) Shows cells of Δ*ygfY* under brightfield and fluorescence. The experiments were repeated three times and similar results were observed.

**Figure 9 microorganisms-09-02316-f009:**
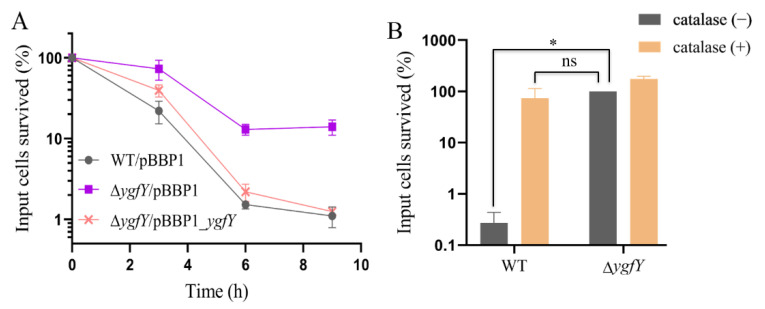
Survival of cells after exposed to heat shock of 42 °C. (**A**) Colony forming units (CFU) of cultures grown on LB plates. (**B**) CFU of cultures incubated in 42 °C for 6 h and then cultured on LB plates with or without catalase overlayed. Error bars indicate standard deviations of results from three biological replicates. Percentage of survived cells are normalized by CFU in cultures before heat exposure (0 h). Asterisk indicates significant difference and ns indicates no significant difference.

**Figure 10 microorganisms-09-02316-f010:**
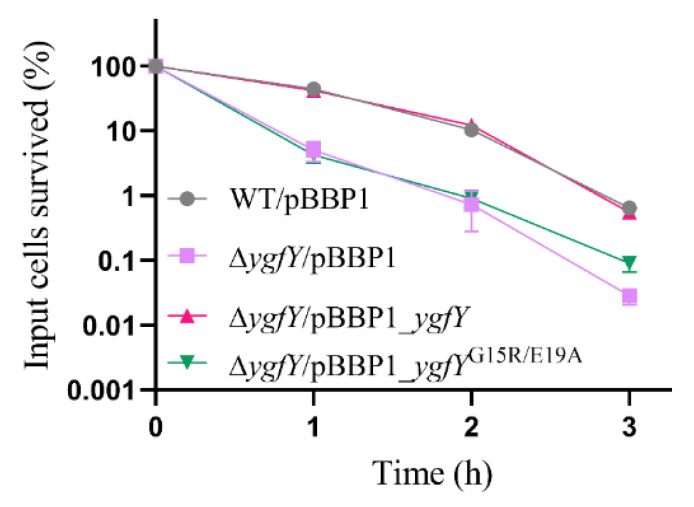
Nitrite tolerance of *S. oneidensis* MR-1. Cultures were exposed to 470 mM of nitrite for indicated time and then spotted on LB plates for cultivation and counting survived cells based on CFU. Percentage of survived cells are normalized by CFU in cultures before nitrite exposure (0 h). The data are the mean ± SD (*n* = 3).

## Data Availability

The data supporting the findings of this study are available within the article.

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
