# Peer review of "YgfY Contributes to Stress Tolerance in Shewanella oneidensis Neither as an Antitoxin Nor as a Flavinylation Factor of Succinate Dehydrogenase"

_microorganisms, 2021, doi:10.3390/microorganisms9112316_

Round 1

Reviewer 1 Report

The manuscript is revised well and I do not have any comments. I agree to publish the revised paper in Microorganisms.

Author Response

Thanks for your time.

Reviewer 2 Report

Line 38-39. TA systems role as stress response modulators is highly controversial (see seminal papers 29895634 and 29233898 and review 31932311) and was studied mostly for type I and II TA systems. Please therefore omit the statement line 38-39

Figure 1. To complete the picture and demonstrate the absence of role of ygfX, please complement ΔygfYX with pBBP1_ygfY. Please also indicate that complementation plasmid drives constitutive expression of ygfY. A cleaner version would be to use the native promoter of ygfY instead of constitutive expression. Please also indicate the copy number of pBBP1 plasmid in the text.

Line 174. The authors might want to rephrase the title to emphasize even better their discoveries. I believe the major discovery here is that (i) ygfXY do not constitute a TA system and (ii) the presumed antitoxin (and not the toxin) plays a role in bacterial growth. It is important to notice that the authors did not show that YfgY positively affect growth, they rather showed that YfgY is required for normal growth.

Lines 239-245. Most proteins become toxic when highly overexpressed as published previously (16672603). This most probably explains the growth defect observed upon overexpression of YgfX as no phenotype is observed in a corresponding deletion mutant and microscopy analyses do not reveal a lemon shaped morphology of the cells. Please discuss this in the text.

Line 250. Please rephrase the title

Figure 5 A-C. The conclusions from this figure are odd and should be revised. The authors believe that the difference in phenotype between ΔygfY and ΔsdhE mutants in the presence of succinate shows that ygfY is not involved in succinate metabolism. However, as the ΔsdhE mutant has no phenotype, and as it is the positive control known to be involved in succinate metabolism, it seems to me that the experimental design is flawed and that one cannot conclude on ygfY function based on these data. In addition, please include a complementation assay for the positive control sdhE as well as for ygfY in figure 5C. Finally, succinate metabolic analysis should not be done in rich medium such as LB. The authors have no evidence that bacteria require succinate metabolism in such conditions and it is most probably not the case knowing how complex and rich LB medium is. Please reproduce the experiment in more defined medium such as MOPS-based medium (4604283) or M9-medium (without casamino acids).

Line 279. Why using the word pleiotropic in the title while you were able to partly delineate the roles of YfgY in bacterial metabolism? Please make the title more precise/informative.

Lines 308-330. It is unclear how a defect in translation capabilities led the authors to test for survival to heat and nitrite stresses. Please elaborate on these missing links in the text to help the reader understand the rationale behind this work.

The authors should discuss the results and significance/novelty of this work in light of the literature: YgfXY homolog in E coli was previously found to form a type II TA system pair (23657679), which could not be confirmed later in Serratia by different group (23657679). Similarly, the authors were unable to reproduce the toxic activity of the toxin in E coli and in Serratia.

Please make sure to harmonize the text with regard to the TA system type. In the introduction, the authors state it is a type IV TA system, while everywhere else, they write type VI.

Author Response

Line 38-39. TA systems role as stress response modulators is highly controversial (see seminal papers 29895634 and 29233898 and review 31932311) and was studied mostly for type I and II TA systems. Please therefore omit the statement line 38-39

The statement is omitted as suggested above.

Figure 1. To complete the picture and demonstrate the absence of role of ygfX, please complement ΔygfYX with pBBP1_ygfY. Please also indicate that complementation plasmid drives constitutive expression of ygfY. A cleaner version would be to use the native promoter of ygfY instead of constitutive expression. Please also indicate the copy number of pBBP1 plasmid in the text.

We have done the suggested experiments. Result of complementing ΔygfYX with pBBP1_ygfY is added in Fig 1B. Result of expressing ygfY from its native promoter is presented as Figure S3.

The pBBP1 is a derivative of broad-host-range cloning vector pBBR1MCS. Based on previous reports, pBBR1MCS has sequences of oriV and rep, and is a low copy number plasmid (PMID: 31197163, 34333160,11222611).

Line 174. The authors might want to rephrase the title to emphasize even better their discoveries. I believe the major discovery here is that (i) ygfXY do not constitute a TA system and (ii) the presumed antitoxin (and not the toxin) plays a role in bacterial growth. It is important to notice that the authors did not show that YfgY positively affect growth, they rather showed that YfgY is required for normal growth.

Thanks for the suggestion. The title is rephrased as “YgfY is required for normal growth of S. oneidensis MR-1”.

Lines 239-245. Most proteins become toxic when highly overexpressed as published previously (16672603). This most probably explains the growth defect observed upon overexpression of YgfX as no phenotype is observed in a corresponding deletion mutant and microscopy analyses do not reveal a lemon shaped morphology of the cells. Please discuss this in the text.

Thanks for the constructive suggestion. And the discussion has been added in the revised manuscript (Line 240-248).

Line 250. Please rephrase the title

The title is rephrased as “The effect of YgfY on growth doesn’t attribute to succinate catabolism”.

Figure 5 A-C. The conclusions from this figure are odd and should be revised. The authors believe that the difference in phenotype between ΔygfY and ΔsdhE mutants in the presence of succinate shows that ygfY is not involved in succinate metabolism. However, as the ΔsdhE mutant has no phenotype, and as it is the positive control known to be involved in succinate metabolism, it seems to me that the experimental design is flawed and that one cannot conclude on ygfY function based on these data. In addition, please include a complementation assay for the positive control sdhE as well as for ygfY in figure 5C. Finally, succinate metabolic analysis should not be done in rich medium such as LB. The authors have no evidence that bacteria require succinate metabolism in such conditions and it is most probably not the case knowing how complex and rich LB medium is. Please reproduce the experiment in more defined medium such as MOPS-based medium (4604283) or M9-medium (without casamino acids).

We totally understand this concern and have examined the succinate consumption by those strains incubated in a mineral medium that is commonly used to culture Shewanella. Our recent result shows that WT is able to consume succinate as reported previously. ΔsdhE shows a defect in succinate consumption compared to WT. In contrast, ΔygfY is comparable to WT in terms of succinate consumption under the tested condition. This result is presented as Fig 5B in the revised manuscript.

Line 279. Why using the word pleiotropic in the title while you were able to partly delineate the roles of YfgY in bacterial metabolism? Please make the title more precise/informative.

The title is rephrased as “YgfY contributes to ability of transcription and translation in S. oneidensis MR-1”.

Lines 308-330. It is unclear how a defect in translation capabilities led the authors to test for survival to heat and nitrite stresses. Please elaborate on these missing links in the text to help the reader understand the rationale behind this work.

This part is rephrased to elaborate on the missing links between translation capacity and heat shock tolerance as follow in the revised manuscript (Line 325-338):

“Then, the translation capacity was compared by examining the tolerance to lethal stresses that cause global protein misfolding. Previous research reports that bactericidal aminoglycosides cause mistranslation, misfolding and aggregation of nascent proteins, which correlates to reactive oxygen species (ROS)A production and consequent cell death  [21, 22]. Inhibitors of protein synthesis, such as chloramphenicol, block ROS-related death of bacterial cells stressed by bactericidal antibiotics [23]. Therefore, we reasoned that if YgfY deficiency causes the lowered capacity of translation, ΔygfY should show an increase in the tolerance to these stresses. The heat shock cause protein misfolding and aggregation [24], and there is a numerical correlation between the protein denature and the death of bacterial cells after the heat shock [25]. Consistent with our presumption, ΔygfY showed an increased tolerance to heat killing than WT (Figure 8A). Moreover, catalase supplementation to quench ROS greatly improved the survival rate of WT, while not so much for that of ΔygfY (Figure 8B). Besides, a weak increase in kanamycin resistance of ΔygfY was observed (data not shown).”

Nitrite tolerance is irrelative to translation capacity.

The authors should discuss the results and significance/novelty of this work in light of the literature: YgfXY homolog in E coli was previously found to form a type II TA system pair (23657679), which could not be confirmed later in Serratia by different group (23657679). Similarly, the authors were unable to reproduce the toxic activity of the toxin in E coli and in Serratia. Please make sure to harmonize the text with regard to the TA system type. In the introduction, the authors state it is a type IV TA system, while everywhere else, they write type VI.

Thanks for these constructive suggestions that are added in the revised manuscript (Line 358-364)

Sorry for typo of TA system, and all those mistakes is corrected.

Round 2

Reviewer 2 Report

  • Results shown at new figure 1B are odd and unexpected. While plasmidic expression of yfgY complements ΔyfgY, it does not complement ΔyfgXY. Such data are inconsistent with growth curves of figure 1A indicating no role for ΔyfgX. Could they authors clarify this point and reproduce growth curves of Figure 1A and 1B? Discussion at lines 390-393 should also be adapted.
  • As suggested before, experiments for figure 5A-B-C should be performed in chemically defined medium and not in LB as bacterial metabolism in LB medium is undefined/complex and therefore not suitable for analysis of specific metabolic pathways.
  • Panels B and D are swapped in figure 5. Could the authors define "mineral medium"? 
  • Lines 347-348. Can the authors show the data for kanamicin resistance? What does "weak increase" mean? Could the authors be more specific and indicate a specific fold increase together with the p value of a statistical test?
  • Line 269, title 3.3: the title is still confusing. Can you rephrase it more clearly?

Author Response

Response to Reviewer 2

  • Results shown at new figure 1B are odd and unexpected. While plasmidic expression of yfgY complements ΔyfgY, it does not complement ΔyfgXY. Such data are inconsistent with growth curves of figure 1A indicating no role for ΔyfgX. Could they authors clarify this point and reproduce growth curves of Figure 1A and 1B? Discussion at lines 390-393 should also be adapted.

Sorry for our mistake. It is the YgfX expression that doesn’t complement ΔyfgXY. Expression of YgfY can complement ΔyfgXY, which is added into Fig 1B in the revised manuscript.

  • As suggested before, experiments for figure 5A-B-C should be performed in chemically defined medium and not in LB as bacterial metabolism in LB medium is undefined/complex and therefore not suitable for analysis of specific metabolic pathways.

Sorry, we haven’t fully understood the reviewer previously. We now replace the Fig 5A as data collected in a defined medium (the mineral medium).

Data in Fig 5B was collected in the mineral medium, which is indicated in the manuscript.

To make the manuscript more logical, Fig 5C and 5D renumber as Fig 6A and 6B.

  • Panels B and D are swapped in figure 5. Could the authors define "mineral medium"? 

There are swapped and renumbered as Fig 6A and 6B.

The mineral medium is composed of 7.956 mM NaCl, 1.93 mM (NH4)2SO4, 0.157 mM MgSO4, 1.29 mM K2HPO4, 1.65 mM KH2PO4, 12.2 mM Na2HPO4, 7.8 mM NaH2PO4, 0.04 mM nitrilotriacetic acid, 34 μM CaCl2, 11.88 μM MnSO4, 3.2 μM CoSO4, 3.13 μM ZnSO4, 1.8 μM FeSO4, 0.526 μM NiCl2, 0.21 μM KAl(SO4)2, 0.2 μM CuSO4. Lactate was added as a carbon source.

  • Lines 347-348. Can the authors show the data for kanamicin resistance? What does "weak increase" mean? Could the authors be more specific and indicate a specific fold increase together with the p value of a statistical test?

In the test of agar diffusion, we observed that the inhibition zone of kanamycin for WT is larger than that for ΔygfY, while, we didn’t quantify the difference. Therefore, the result about kanamycin is deleted in the revised manuscript for the scientific rigor.

  • Line 269, title 3.3: the title is still confusing. Can you rephrase it more clearly?

The title is rephrased as “YgfY doesn’t attribute to succinate catabolism” in the revised manuscript.

This manuscript is a resubmission of an earlier submission. The following is a list of the peer review reports and author responses from that submission.

Round 1

Reviewer 1 Report

Microorganisms

(1374359)

The manuscript "YgfY contributes to stress tolerance in Shewanella oneidensis 2 neither as an antitoxin nor as a flavinylation factor of succinate dehydrogenase" by Zhang et al. describes about YgfX importance under some stress condition in Shewanella oneidensis. The authors found that the delation of YgfY but not YgfX caused cell growth arrest at low temperature. They also showed that transcription of ribosomal proteins was significantly repressed in the mutant. However, the most critical point, the function of YgfY was not identified and the manuscript does not enhance our understanding of this phenomenon and more in-depth studies are required to warrant publication in Microorganisms.

Major Comments

  1. Page 7 (Figure 6 and Table S3), authors described that the genes encoding ribosomal proteins were down-regulated in ∆ygfY mutant. Were other house-keeping genes analyzed? In my knowledge, many genes containing ribsomal proteins were transcribed with sigma70. Why were only ribosomal genes repressed? The authors must analyze the transcrption rate and/or RNA polymerase activity in ygfY deletion mutant to predict the function of YgfY.

Minor Comments

Numerous errors were found throughout the manuscript,.

  1. Page 1, lanes 37, “t” should be removed.

  2. Page 1, lanes 44, “six TA systems” should be ”six type of TA systems”

  3. Page 4, lanes 173, The data should be shown in supplementary data.

  4. Page 5, lanes 196. Is E. coli BL21 used in this study? How was the protein expressed from pET28a without DE3?

  5. Page 5, lanes 203-206. As mention, overexpression of some proteins affect on cell growth. And it was also known that overexpression of His6-tag cause growth arrest. Can we tell if the growth defect is caused by overexpression of proteins or specific toxicity?

  6. Page 7, lanes 251, The YgfYG16RE19A mutant was expressed as soluble protein?

Reviewer 2 Report

In this manuscript Zhang M-X. et al., propose a nobel mechanism of YgfY function in stress tolerance of Shewanella species, The authors have demonstrated that YgfY has pleiotropic impacts in S. oneidensis MR-1 as adaptative groth at low temperature & nitrite tolerance but they don´t attributed then to antitoxin and flavinilation factor of SDS functions.

The experiments has been correctly performed and support the conclusions of the authors.

Some minor comments

Line 37: tolerance in the Shewanella species

Line 120: 16 ºC

Figure 1B: Error bars are not shown in the graph.

Line 195: Hetero-expression 

Line 264: nsrR is downregulated at ∆ygfY. Have you analyzed the transcription of this gene by RNA-Seq or qRT-PCR? The results are the same as those obtained with the nrfA analysis